# LCMR1 Promotes Large-Cell Lung Cancer Proliferation and Metastasis by Downregulating HLA-Encoding Genes

**DOI:** 10.3390/cancers15225445

**Published:** 2023-11-16

**Authors:** Lu Liu, Chunsun Li, Zhen Wu, Yanqin Li, Hang Yu, Tao Li, Yueming Wang, Wei Zhao, Liangan Chen

**Affiliations:** 1Medical School of Chinese PLA, Beijing 100853, China; deer_3016@163.com (L.L.); yuhang_301@126.com (H.Y.); drtaolee@outlook.com (T.L.); 2Department of Nutrition, The First Medical Center of Chinese PLA General Hospital, Beijing 100853, China; 3Department of Pulmonary and Critical Care Medicine, Chinese PLA General Hospital, Beijing 100853, China; sun082000@163.com (C.L.); wuzhen08261991@163.com (Z.W.); lyq.1980@163.com (Y.L.); wangym931227@163.com (Y.W.); 4Department of Oncology, The First Medical Center of Chinese PLA General Hospital, Beijing 100853, China; 5School of Medicine, Nankai University, Tianjin 300071, China

**Keywords:** NSCLC, proliferation, migration, LCMR1, human leukocyte antigen

## Abstract

**Simple Summary:**

LCMR1 is a subunit of the mediator complex, also known as MED19, which is involved in various life activities and is closely related to the occurrence and development of tumors. In this study, we knocked down LCMR1 in large cell lung cancer cells and found that high expression of LCMR1 in LCLC indicates a poor prognosis. Meanwhile, blocking LCMR1 in the 95D LCLC cell line reduced proliferation and metastasis in vitro and in living organisms. LCMR1 inhibits the transcription of HLAs, a gating factor of cancer-specific antigen-presenting.

**Abstract:**

Lung cancer is notorious for its high global morbidity and mortality. Here, we examined whether the *LCMR1* gene, which we previously cloned from a human large-cell lung carcinoma cell line, contributes to the proliferation and metastasis of large-cell lung carcinoma. To this end, we performed pan-cancer and non-small cell lung cancer (NSCLC) cell line-based LCMR1 expression profiling. Results revealed that LCMR1 was expressed at high levels in most solid tumors, including NSCLC. LCMR1 expression was the highest in the 95D large cell lung cancer cell line. Functional studies using lentivirus-based knockdown revealed that LCMR1 was critical for the proliferation, migration, and invasion of cultured large cell lung cancer cells. Moreover, blocking this gene significantly reduced tumor growth in a 95D cell xenograft mouse model. A multiple sequence-based assay revealed a mechanism by which LCMR1 diminished the RNA Pol II occupancy at the promoter of human leukocyte antigen (HLA)-encoding genes to prevent their transcription. The HLA genes play vital roles in cancer-specific antigen presentation and anticancer immunity. A correlation assay using TCGA database identified a negative relationship between the expression levels of *LCMR1* and HLA coding genes. Taken together, our findings demonstrate that LCMR1 is required for large cell lung cancer cell growth and invasion and suggest its potential as a valid target in clinical treatment.

## 1. Introduction

Lung cancer is one of the most frequently diagnosed malignancies and continues to be the primary contributor to cancer-related fatalities globally, affecting individuals of all genders [1]. The major histological subtypes of lung cancer encompass small cell lung cancer and non-small cell lung cancer (NSCLC). The latter is responsible for more than 1.38 million global deaths annually, accounting for approximately 85% of lung cancer cases [2]. NSCLC can be further categorized into adenocarcinoma, squamous cell carcinoma, and large cell lung carcinoma (LCLC). Identifying specific molecular alterations within the different forms of lung cancer has changed how clinicians treat this disease [3]. Despite remarkable advancements in NSCLC treatment, the overall cure and survival rates, especially in metastatic cases, remain suboptimal. Consequently, the molecular mechanisms underlying the progression of NSCLC need further exploration to improve NSCLC treatment.

We previously used a differential display polymerase chain reaction technique to clone a novel lung cancer metastasis-related protein 1 (*LCMR1*) gene from a poorly differentiated human LCLC cell line [4]. *LCMR1*, situated on human chromosomal locus 11q12.1, comprises 949 nucleotides with an open reading frame (ORF) encoding a 177 amino acid peptide in the human genome. *LCMR1* is also recognized as mediator complex subunit 19 (*MED19*), a constituent of the mediator complex that facilitates transcription activation by linking transcription factors and RNA polymerase II [5]. Although *LCMR1* was first cloned in LCLC cells, many reports have since revealed that it regulates the proliferation and migration of breast cancer [6], prostate cancer [7], gastric cancer [8], and tongue cancer [9] cells. In lung adenocarcinoma cells, knocking down LCMR1 reduces cell proliferation and tumorigenesis [10,11,12]; yet its role in LCLC, where it was originally identified, remains to be established [13,14].

In this study, we examined the expression of *LCMR1* in 95C and 95D cells derived from a poorly differentiated human LCLC cell line, exhibiting low (95C) or high (95D) metastatic capacity. We created a lentiviral vector that specifically targets *LCMR1* and examined the impact of inhibiting *LCMR1* expression in 95D cells on the growth and migration of cancer cells in vitro and in vivo. We also explored the mechanism underlying the effect of LCMR1 by performing RNA-seq and RNA-poll II ChIP-seq assays. Our analyses revealed that LCMR1 promotes cancer metastasis by blocking the transcription of human leukocyte antigen (HLA)-coding genes. Therefore, the knockdown of *LCRM1* may offer a rational clinical approach for treating LCLC.

## 2. Materials and Methods

### 2.1. Cell Lines and Culturing

Dr. Lezhen Chen (Department of Pathology, Chinese PLA General Hospital, China) generously provided the 95C and 95D cell lines derived from a poorly differentiated human LCLC cell line, PLA-801. ShangHai Biowing Applied Biotechnology Co., Ltd. (Shanghai, China) authenticated the cell lines through STR analysis, following the guidelines of Capes-Davis21 and the ANSI Standard (ASN-0002) established by the ATCC Standards Development Organization. ATCC supplied the remaining cell lines, namely H292, A549, H1975, H446, and H1688. The cells were grown in RPMI-1640 medium supplemented with 10% fetal bovine serum, penicillin (100 μg/mL), and streptomycin (100 μg/mL). The cells were cultured at 37 °C in a moist environment with 5% CO_2_.

### 2.2. Delivery of Short Hairpin (sh) RNA Using Lentivirus

We created five shRNA sequences targeting LCMR1 and integrated them into a lentivirus RNA expression system (pGCSIL-GFP) provided by Vigen Co., Ltd. (Zhenjiang, China). The effective targeting sequence of LCMR1 (CAGTAGCTCTTTCAATCCTAT) was selected by immunoblotting. A sh-NC—a shRNA sequence that does not cause silencing—was employed as a negative control. Vigen Co., Ltd. (Zhenjiang, China) performed the additional shRNA vector transfection and lentiviral packaging. The 95D cells were placed in 6-well dishes, cultivated until reaching a 70–80% density, and exposed to sh-LCMR1 or sh-NC lentivirus using 8 μg/mL polybrene (Sigma-Aldrich, St. Louis, MO, USA). Stably transfected cell lines were selected using GFP fluorescence as a sorting marker. Cells with >75% infection efficiency were used in further analyses.

### 2.3. Immunofluorescence Staining

To perform immunofluorescence staining, the 95D cells were treated with 4% paraformaldehyde and fixed at room temperature for 20 min. Following three rounds of washing with phosphate-buffered saline (PBS), the cells were disrupted using 0.1% Triton X-100 in PBS for 30 min. Subsequently, they were subjected to blocking with 1% BSA at 37 °C for 1 h. The cells were cultured overnight with a rabbit anti-MED19 (#PA5-44383, Invitrogen, Life Technologies, Carlsbad, CA, USA) antibody at 4 °C. Following three rinses with PBS, the cells were exposed to the respective secondary antibodies for 1 h at 37 °C. The cells were stained with DAPI (4′, 6-diamidino-2-phenylindole) for 2 min, followed by a PBS wash. Images were acquired with an Olympus confocal microscope and edited using Olympus FV1000 software (FV10-ASW, version 4.2).

### 2.4. Proliferation Assay

The Cell Counting Kit-8 (CCK-8; Beyotime, Shanghai, China) was used to assess cell proliferation. Following 2 days of transfection, 5 × 10^3^ cells were seeded per well in a 96-well plate and grown for 24, 48, 72, 96, and 120 h. Prior to analysis, cells were treated with 10 µL CCK-8 and 90 µL medium at 37 °C for 1 h; their light absorbance was measured at a wavelength of 450 nm.

### 2.5. Migration Assays

#### 2.5.1. Wound Healing Assay

A wound-healing experiment was conducted in accordance with a previously provided protocol. The cells were seeded in 6-well dishes and incubated overnight. Injuries were inflicted using a 200 μL pipette tip. Images were obtained of the wound site after 0 and 24 h. The extent of wound healing was measured with the ImageJ program and standardized based on the initial width of the wound (0% closure, 0 h).

#### 2.5.2. Transwell Migration and Invasion Assay

Transwell migration assays were executed using plates with 0.8 µm pore polycarbonate membranes (Corning, NY, USA), as described previously [15]. After being seeded in the upper chamber, cells were cultured for 12 and 24 h. Subsequently, the cells on the upper side were removed using cotton swabs, while the invaded cells that penetrated the membrane were fixed with methanol and 1% crystal violet for 10 min before being visualized under three different fields. After washing the crystal violet with a 33% solution of acetic acid, the absorbance at 570 nm was measured. Migration activity was quantified based on the OD value.

### 2.6. Cell Cycle Analysis

After lentivirus infection, cells were trypsinized to obtain single-cell suspensions. These suspensions were collected and washed with DPBS (Solarbio, Beijing, China). Subsequently, the cells were fixed in 70% ethanol and incubated at −20 °C overnight. After centrifugation, the ethanol was removed, and the cell pellets were rinsed with DPBS. The cells were incubated with 100 μL of propidium iodide (PI, KeyGEN, Nanjing, China) containing RNase A for 5–10 min at 37 °C in the absence of light. Flow cytometry (Beckman Coulter, Brea, CA, USA) was used to assess the cell cycle status. ModFit LT 5 software calculated the cell ratio in each cycle.

### 2.7. In Vivo Animal Assays

This study was approved by the Institutional Animal Care Use Committee of the Chinese People’s Liberation Army General Hospital (No. 2022-X18-40). The in vivo impact of LCMR1 on LCLC growth was examined using 4-week-old male BALB/C nude mice, randomly separated into two groups (*n* = 6 per group). 95D cells transfected with stable sh-LCMR1 (sh-RNA group) or sh-NC (control group) were injected subcutaneously into the flank region of the legs at a concentration of 5 × 10^6^ cells per mouse. The tumor’s dimensions were determined by measuring the lengths of the perpendicular axes and calculating the volume using the formula: volume = (length × width²)/2. After 45 days, the mice were euthanized through dislocation of the cervical vertebrae. The primary tumors were collected and weighed.

### 2.8. Protein Extraction and Immunoblotting

The cultured cells were collected, lysed using RIPA lysis solution, and centrifuged to determine the protein concentration in the resulting liquid. After loading protein samples (40 μg) onto SDS-PAGE gels, they were electrophoresed and transferred to PVDF membranes. Following membrane blocking using 5% skim milk for 1 h at ambient temperature, the blots were incubated with primary antibodies overnight at 4 °C. The bound primary antibodies were treated with the corresponding secondary antibodies (1:5000, cat. no. ZB-2301, ZSGB-BIO; 1:5000, cat. no. ZB-2305, ZSGB-BIO) overnight at 4 °C and detected using the enhanced chemiluminescence reagent (APPLYGEN, Beijing, China). The primary antibodies used were (dilution 1:1000) anti-MED19 (#PA5-78656, Invitrogen) and anti-β-Actin (#20536-1-AP, Proteintech, Wuhan, China).

### 2.9. RNA Extraction and Quantitative PCR

Trizol reagent (Invitrogen) was used to extract total RNA from cultured 95D cells. Following precipitation with isopropanol and subsequent rinsing with 75% ethanol, the RNA was quantified by calculating the A260/A280 absorbance ratio utilizing a Nanodrop spectrophotometer (Thermo Fisher Scientific, Waltham, MA, USA). A PrimeScript RT reagent Kit with a gDNA Eraser (Takara, RR047A) was employed for reverse transcription using a 1 μg RNA sample. The KAPA SYBR FAST Universal kit (KAPA Biosystems, KK4601, Cape Town, South Africa) was used for quantitative PCR on a MyiQ2 Two-Color Real-Time PCR Detection System (Bio-Rad, Hercules, CA, USA). To determine the quantitative expression levels of the target genes, the 2^−ΔΔCt^ method was employed. The amplification primers are provided in Appendix A. The quantity of cDNA was normalized based on the β-Actin housekeeping gene.

### 2.10. Transcriptome Sequencing and Data Analysis

To perform RNA-seq, we followed the aforementioned procedure to collect total RNA. Then, 1 μg of RNA was utilized to prepare the RNA-seq library and conduct sequencing on an Illumina HiSeq X Ten system (Annoroad Gene Technology Co., Ltd., Beijing, China).

For RNA-seq data processing, clean reads were aligned to the GRch38 genome using tophat2 [16]. Only reads that were uniquely mapped were utilized for gene read number counting, while the expression abundance of each assembled transcript was measured using fragments per kilobase of exon model per million (FPKM) mapped read values. To screen differentially expressed genes (DEGs), the edgeR R Bioconductor package [17] was utilized. A false discovery rate of <0.05 and fold change >2 or <0.5 were set as the cut-off criteria for identifying DEGs. Functional categorization of gene ontology (GO) [18] terms was conducted based on molecular function, biological process, and cellular component ontologies, using an E-value threshold of 10^–5^. We acquired extra pathway enrichment data from the Kyoto Encyclopedia of Genes and Genomes (KEGG) [19].

### 2.11. ChIP-Seq and Data Processing

The CUT&Tag experiment was conducted according to previous instructions, with a few alterations [20]. In short, native nuclei were isolated from 95D cells and gently washed twice with a wash buffer containing 20 mM HEPES pH 7.5, 150 mM NaCl, 0.5 mM spermidine, and 1× protease inhibitor cocktail. Beijing Novogene Co., Ltd (Beijing, China). carried out tagmentation, library construction, and sequencing.

To process the sequencing reads, we utilized the previously described bioinformatic pipeline [21]. Prior to mapping pair-end reads, clean reads were acquired from the original reads by eliminating the adapter sequences utilizing the trimmomatic software (Version 0.36) [22]. The pristine reads were mapped to the mm10 genome sequences using the BWA software (Version 0.7.17) [23]. We used macs2 (Version 2.2.8) [24] to call the peaks and selected peaks with a cutoff q value < 0.05. The bam file produced by the uniquely mapped reads was utilized as an input with deeptools software (Version 3.2.1) [25] to generate bigwig files. The deeptools software (Version 3.2.1) is used to visualize read distribution (from bigwig files) over peaks; the peaks are annotated using the annotatePeak feature of ChIPseeker [26]. The HOMER’s [27] tool was used for motif analysis, while the IGV tool was employed for visualization.

### 2.12. Data Availability

The unprocessed datasets and the examined bigwig and narrowpeak files produced in the present investigation can be found in the GEO database GSE 234816.

### 2.13. Pan-Cancer Analysis

Using the TIMER2.0 database [28], the ‘Gene_DE-Exploration’ function analyzed and compared the expression level of LCMR1 in tumors and matched normal tissues across all TCGA cancer types.

We conducted a correlation assay between LCMR1 and the HLA-encoding genes across TCGA, focusing on NSCLC, using the “Gene_Corr-Exploration” function of the TIMER2.0 database.

The default parameters were used in the above analyses.

### 2.14. Statistical Analysis

The data were expressed as mean ± SEM. To evaluate significant distinctions, a two-tailed Student’s *t*-test or one-way ANOVA was employed to compare multiple groups. GraphPad Prism 8 (GraphPad Software) was utilized for all data analyses. A significance level of <0.05 was deemed statistically significant.

## 3. Results

### 3.1. LCMR1 Is Highly Expressed in LCLC Cell Lines

We first analyzed the expression of LCMR1 (MED19) across various TCGA tumors. Profiling data showed that LCMR1 was mainly upregulated in tumor tissues, including NSCLC, such as lung adenocarcinoma (LUAD) and lung squamous cell carcinoma (LUSC) (Figure 1A). We then analyzed the RNA and protein expression levels of LCMR1 in different NSCLC cell lines and compared them to the normal BEAS-2B lung epithelial cell line. The 95D cell line—an LCLC subclone with high invasive capacity [4]—exhibited the highest mRNA levels of LCMR1 (~5-fold compared with BEAS-2B), whereas the 95C LCLC cell line, with low metastatic potential, as well as the other lung cell cancer lines (H292, A549, H1975, and H446) showed lower LCMR1 expression (~2- to 3-fold changes) (Figure 1B). The protein-level profiling showed the same results, with the highest level in the 95D cells (Figure 1C). Immunostaining for LCMR1 in 95D cells showed nuclear localization (Figure 1D), which is consistent with its bio-function as a transcriptional mediator [29].

We also used lentivirus-based RNA interference to determine whether LCMR1 participates in LCLC growth and metastasis. Of the five generated viruses, sh-a and sh-e effectively reduced LCMR1 mRNA and protein levels in 95D cells (Figure 1E,F). The sh-a strain was used for subsequent assays as it was more effective at reducing RNA levels.

### 3.2. LCMR1 Contributes to LCLC Growth and Invasion In Vitro

We examined the effect of LCMR1 expression on cell growth rate using two different comparisons: (1) the 95C cells with naturally low LCMR1 expression versus the 95D cell line with high LCRM1 expression; (2) 95D cells transfected with control virus (95D-NC) versus 95D cells transfected with LCMR1-interfering virus (95D-shRNA). The proliferation rates were higher for the 95D cells than for 95C cells (Figure 2A, left). Similarly, knocking down LCMR1 in 95D significantly reduced the proliferation rate (Figure 2A, right).

We previously showed that LCMR1 was strongly overexpressed in NSCLC and that its expression positively correlated with cell migration and invasion [4]. Comparisons of the cell adhesion and wound-healing assays showed similar results to the proliferation findings: the cell adhesion rate (Figure 2B,C; left) differed only slightly (not statistically significant) between the 95D and 95C cells; however, knocking down LCMR1 in 95D cells strongly reduced cell adhesion (Figure 2B,C; right). The trans-well chamber assay results showed greater invasiveness for the 95D cells than 95C cells (Figure 2D left), while knockdown LCMR1 significantly reduced the invasiveness of the 95D cells (Figure 2D right). Taken together, these loss-of-function studies demonstrated that LCMR1 positively regulates the proliferation and invasion of LCLC cells.

The flow cytometry results used to detect the cell cycle distribution of the 95C and 95D cells revealed fewer 95D than 95C cells in the S phase, while the knockdown of LCRM1 in 95D cells resulted in a more severe S phase arrest (Figure 2E). The quantitative analysis identified more cells in the S phase (Figure 2F), suggesting that *LCMR1* knockdown induces S-phase arrest.

### 3.3. Knocking down LCMR1 Inhibits In Vivo Tumor Growth of 95D Xenografts Model

Investigation of nude mice bearing 95D-NC or 95D-shRNA tumor xenografts revealed no change in body weight during the recording period (Figure 3A). The in vivo tumor growth of 95D cell xenografts was significantly reduced by LCMR1 knockdown (Figure 3B). The calculated tumor volume (Figure 3C) and weight (Figure 3D) were much lower for 95D-shRNA cell tumors than for 95D-NC tumors. The knockdown efficiency was confirmed by analyzing the expression of LCMRl in tumor samples via immunoblot (Figure 3E). The above data suggested that LCMR1 contributes to the LCLC tumor growth in vivo.

### 3.4. LCMR1 Negatively Regulates Human Leukocyte Antigen (HLA) Coding Genes

Transcriptome sequencing using sh-LCMR1 or sh-NC lentivirus-infected 95D cells and construction of a correlation matrix based on the calculated fragments per kilobase of transcript per million fragments mapped (FPKM) revealed higher correlations within the biological replicates in the same groups, indicating a consistent transcriptional difference (Figure 4A).

A standard fold change (FC) ≥2 or ≤−2 and a false discovery rate (FDR) ≤0.05 were used to screen the differentially expressed genes (DEGs), identifying 176 significantly upregulated genes and 39 downregulated genes. The Kyoto Encyclopedia of Genes and Genomes (KEGG) pathway analysis of the upregulated and downregulated genes showed that the homologous recombination, DNA replication, mismatch repair, proteasome, and base excision repair pathways were suppressed, while the autoimmune thyroid disease, neuroactive ligand-receptor interaction, and olfactory transduction pathways were activated (Figure 4B). The heatmap of genes enriched in these pathways showed that the changes in the downregulated pathways were not statistically significant, while the activated pathways showed significant changes (Figure 4C). A gene set enrichment analysis (GSEA) demonstrated that LCMR1 depletion activated the autoimmune thyroid disease pathway (Figure 4D).

Many HLA-encoding genes, characteristic of autoimmune thyroid disease, were upregulated in the LCMR1-shRNA group (Figure 4C), indicating they could be downstream targets. HLA presents intra-cellular peptides on the cell surface for recognition by T-cell receptors. Hence, it is a critical step in cancer-specific antigen presentation and subsequent T-cell activation. Loss of class-I HLA expression could lead to immune escape and is positively associated with NSCLC progression [30,31]. The transcriptional regulation effect of LCMR1 on HLA-encoding genes was confirmed as an LCMR1 loss of function in 95D cells. The mRNA levels of HLAa, HLAb, HLAc, and many HLAd members were higher in 95D-shRNA cells (Figure 4E) than in 95D-NC control cells.

### 3.5. LCMR1 Diminishes the RNA Pol II Occupancy at the Promoter of HLA-Encoding Genes

Pol II ChIP-seq performed using sh-LCMR1 or sh-NC lentivirus-infected 95D cells revealed good sequencing depth and quality; 19,413 peaks were called in the 95D-NC samples, and 19,757 peaks in the 95D-shRNA samples (Figure 5A). The peak distribution showed Pol II enrichment at the ±1 kb promoter region of all target genes (Figure 5B), consistent with a transcription mode of action. Annotation and analysis of the target gene of each peak revealed higher normalized read counts for the LCMR1-shRNA cells than for the control cells (Figure 5C,D), indicating that blocking LCMR1 facilitates the binding of RNA Pol II onto promoters. Among all sequencing peaks, 5694 were specific to the control group, 6374 were specific to the treatment group, and 13,068 were in both groups (Figure 5E).

Knocking down *LCMR1* dramatically increased the RNA Pol II occupancy on all Class I HLA genes, including *HLA-A*, *HLA-B*, and *HLA-C* (Figure 5F left), and parts of the Class II HLA genes, such as *HLA-DOB*, *HLA-DMB*, and *HLA-F* (Figure 5F right). The ChIP-seq results corresponded to the previous qPCR assay findings, indicating that LCRM1 blocked the transcription of HLA-encoding genes by diminishing the RNA Pol II occupancy on the promoter region.

### 3.6. The Expression Level of LCMR1 Negatively Correlates with HLA-Encoding Genes in NSCLC Samples

Evaluation of the association between *LCMR1* expression and HLA levels using the Timer2.0 database [32] revealed that LCMR1 was negatively correlated with HLA-encoding genes in NSCLC samples (Figure 6A). The specific scattergrams for the linear correlations between LCMR1 and each HLA-encoding gene further confirmed the negative correlation between these genes (Figure 6B), suggesting a significant relationship between LCMR1 and HLA expression in clinical NSCLC samples.

## 4. Discussion

Due to the absence of early identification methods and the acknowledgment of symptoms at an advanced stage, the five-year survival rate for patients with NSCLC is significantly lower (17.8%) than for other prevalent forms of cancer. Treatment failure in NSCLC is primarily caused by metastasis, which is why clinical intervention involves molecularly targeted drugs, palliative radiotherapy, and immune checkpoint inhibitor treatment. LCLC [33] is a relatively rare and aggressive type of undifferentiated NSCLC, occurring in less than 1% of all lung cancer surgical specimens [34]. Compared to other subtypes of NSCLC, LCLC is more aggressive and has a poorer prognosis [35,36]. This study investigated the potential function of *LCMR1*, a gene that we previously discovered to show a strong correlation with LCLC malignancy.

In line with our and our colleagues’ previous discoveries, LCMR1 is positively associated with the proliferative ability of human NSCLC cells [10,11,12,13]. Here, we showed that LCMR1 can suppress the in vitro and in vivo proliferation of 95D cells—a subcloned cell line derived from a poorly differentiated human LCLC. Knocking down *LCMR1* reduced cell adhesion and migration and led to cell cycle arrest at the S phase. Transcription sequencing and RNA Pol II ChIP-seq results suggested that *LCMR1* expression diminished RNA Pol II occupancy at the promoter of HLA-encoding genes. We found that LCMR1 negatively regulates the transcription of these genes in cultured cells and clinical samples. Given the crucial role of HLA in cancer-specific antigen presentation and subsequent T-cell activation, our findings suggest that the high expression of *LCMR1* in LCLC could lead to immune checkpoint escape by inhibiting HLA expression, ultimately leading to poor prognosis.

Regarding the relationship between the opposing characteristics of adhesion and floating and tumor metastasis, there are divergent opinions. Most scholars believe that floating cells have more metastatic features than adherent cells, and a decrease in cell adhesion usually increases the metastatic ability of cells. However, tumor metastasis is a complex process influenced by various factors, including intracellular and intercellular interactions, the tumor microenvironment, and the immune system. Specific situations or mechanisms may lead to a decrease in cell adhesion while also reducing the migration ability of cells [37]. In addition, certain treatment methods may decrease cell adhesion and metastasis ability [38]. In our study, blocking LCMR1 decreased cell proliferation and reduced the invasion activity as measured by trans-well assay while increasing the floating cells in the adhesion assay. Taken together, our data revealed that blocking LCMR1 reduced the metastatic features. The precise regulating effect of LCMR1 on invasion will be explored in future in vivo assays.

Mechanistically, we found that LCMR1 can affect the binding of RNA Pol II to the promoter regions of HLA-encoding genes and thus inhibit their transcription. The class I HLAs, such as *HLA-A*, *HLA-B*, and *HLA-C*, play vital roles in anticancer immunity: HLA initiates anticancer immunity by presenting antigenic peptides on the cell surface. These are recognized by CD8^+^ T-cells, and the tumor cells are killed [39]. Consequently, any downregulation of HLA presentation could provide an attractive escape mechanism for lung cancer cells, allowing them to avoid immune recognition [31]. Interestingly, LCMR1 (MED19) reportedly down-regulates C-X-C motif chemokine ligand 11 (CXCL11) in breast cancer; blocking LCMR1 contributes to high CXCL11 levels and positively correlates with antitumor immune responses [40]. Our findings, therefore, reveal an additional mechanism by which the increased expression of *LCMR1* inhibits HLA transcription, blocking antigen presentation and subsequent anti-tumor immune responses. This mechanism is consistent with our previous clinical studies, in which we found a significant overexpression of *LCMR1* in human LCLC and a close positive association between its expression and the clinical stage of patients. We will explore the regulatory effect of LCMR1 on cancer immunity in future experiments and elucidate the contribution of HLA-based antigen presentation to this process.

## 5. Conclusions

Taken together, our findings indicate that *LCMR1* expression can directly reduce HLA-encoding gene transcription and promote LCLC cell proliferation and metastasis. Thus, the inhibition of LCMR1 may be a key strategy for activating anticancer immunity and treating lung cancer.

## Figures and Tables

**Figure 1 cancers-15-05445-f001:**
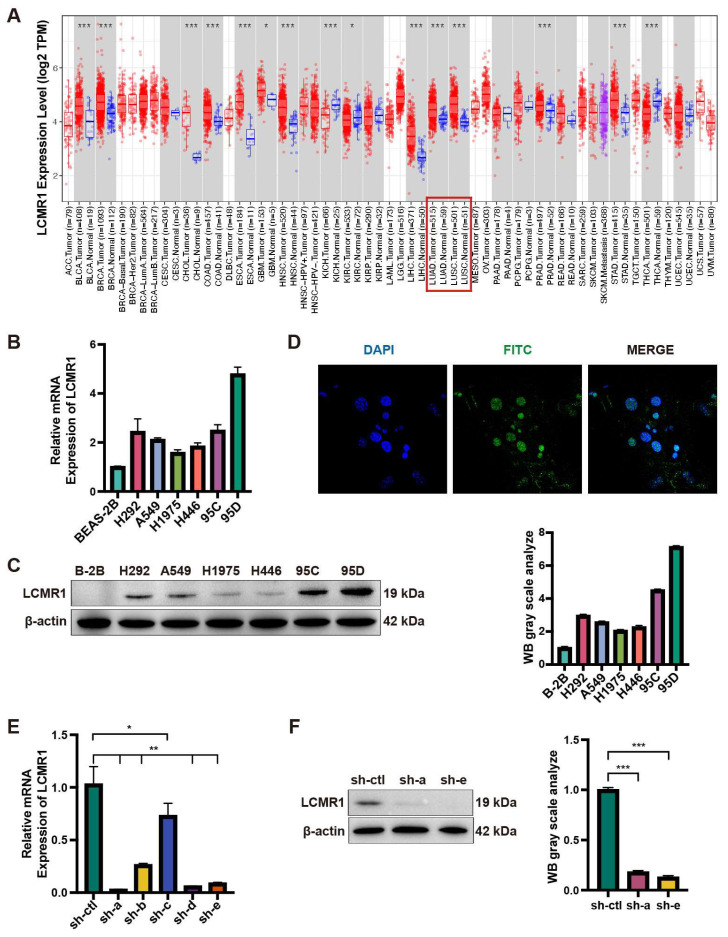
Expression level of LCMR1 in NSCLC. (**A**) Aberrant mRNA expression of LCMR1 in pan-cancer from the TIMER2.0 database indicated LCMR1 expression in 18 cancer types. The red and blue boxes represent tumor tissues and normal tissues, respectively. The lung adenocarcinoma (LUAD) and lung squamous cell carcinoma (LUSC) are indicated with red box. (**B**,**C**) The LCLC cell lines (95C and 95D) and adenocarcinoma cell lines (H292, A549, H1975, and H446) were used to profile the expression of LCMR1 in NSCLC cells. The relative mRNA levels of these cell lines were analyzed by quantitative PCR and normalized to the normal lung epithelial cell BEAS-2B (**B**). The protein levels were analyzed by western blotting (The uncropped blots are shown in Appendix A) (**C**). (**D**) Immunofluorescence staining of LCMR1 in 95D cells, showing subcellular nuclear localization. Blue, DAPI; Green, LCMR1; Cyan, merge. (**E**,**F**) The 95D cells were transfected with shRNA constructs to knock down endogenous LCMR1 expression. (**E**) LCMR1 mRNA levels of the five designed lentivirus-infected 95D cells were tested by qPCR and compared with control virus-infected cells. (**F**) Left: western blot analysis of the knockdown effect of the a- and e-lines of the virus in 95D cells (The uncropped blots are shown in Appendix A); Right: arbitrary unit of immunoblot. sh-a strain was used for subsequent detection and sequencing. Results represent the mean and standard deviation of three independent experiments. Student’s *t*-test * *p* < 0.05; ** *p* < 0.01, *** *p* < 0.001.

**Figure 2 cancers-15-05445-f002:**
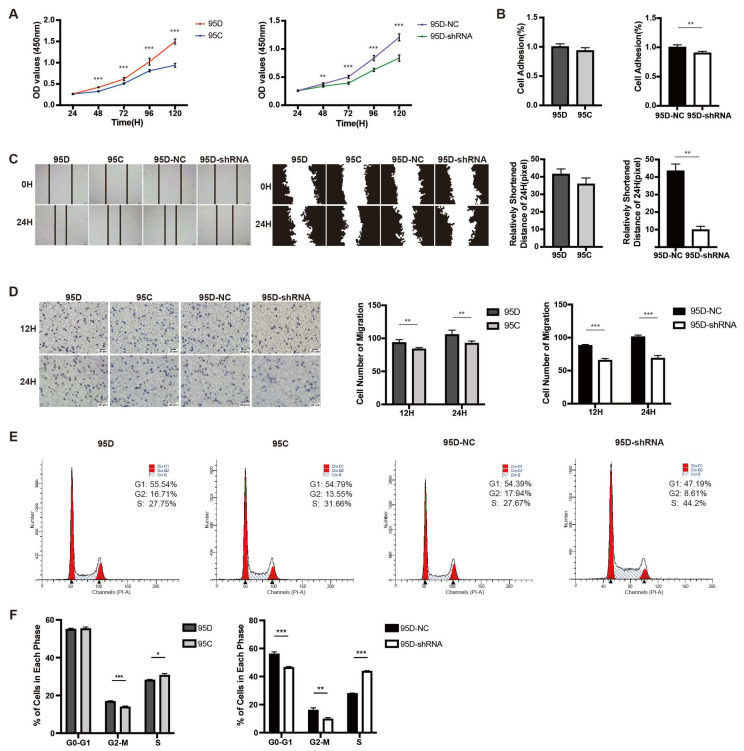
Effect of LCMR1 on proliferation, adhesion, migration, and cell cycle. The wildtype 95D cells were compared with wildtype 95C cells, while the control virus-infected 95D cells were compared with LCMR1-shRNA virus-infected 95D cells. (**A**) CCK-8 assay of the above-mentioned cells at different times after passage. (**B**) Adhesion ability assay. (**C**) Wound healing assay for motility. Left: photos of the wound healing assay. Middle: gray image of cell edges. Right: calculated closure rate of migrating cells at 24 h versus 0 h. (**D**) The above-mentioned cells were inoculated in Trans-well chambers for 12 or 24 h to assess migration with crystal violet staining. Left: representative photomicrographs of the cell staining. Right: Statistical analysis of the stained cell numbers. (**E**,**F**) Effect of LCMR1 on cell cycle distribution in the above-mentioned cells. (**E**) Cells assessed by flow cytometry; (**F**) Percentage of G0-G1, G2-M, and S fraction populations plotted in a histogram. Results represent the mean and standard deviation of three independent experiments. Student’s *t*-test * *p* < 0.05; ** *p* < 0.01, *** *p* < 0.001.

**Figure 3 cancers-15-05445-f003:**
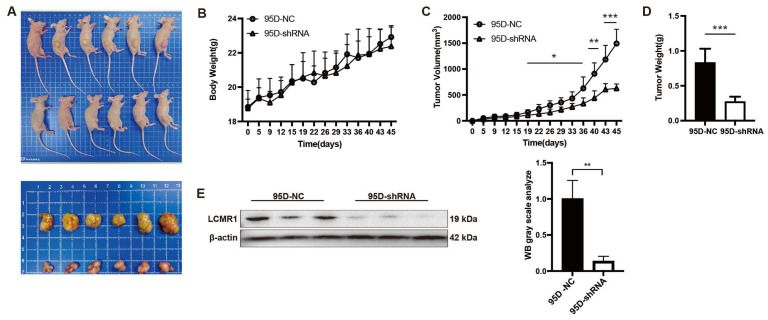
Blocking LCMR1 inhibits transplanted tumor growth in vivo. Mice were inoculated subcutaneously into the right flank with either control- or LCMR1-shRNA virus-infected 95D cells (*n* = 6 per group) at a dosage of 5 × 10^6^ cells per mouse. (**A**) Inset images show the sizes of representative tumors. (**B**) Body weight change in 95D tumor-bearing mice. (**C**) Tumor growth curve of 95D tumor-bearing mice. (**D**) Tumor weight. (**E**) Left: western blot analysis of the LCMR1 expression with tumor tissues (The uncropped blots are shown in Appendix A); Right: arbitrary unit of immunoblot. Results represent the mean and standard deviation of two independent experiments. Student’s *t*-test * *p* < 0.05; ** *p* < 0.01, *** *p* < 0.001.

**Figure 4 cancers-15-05445-f004:**
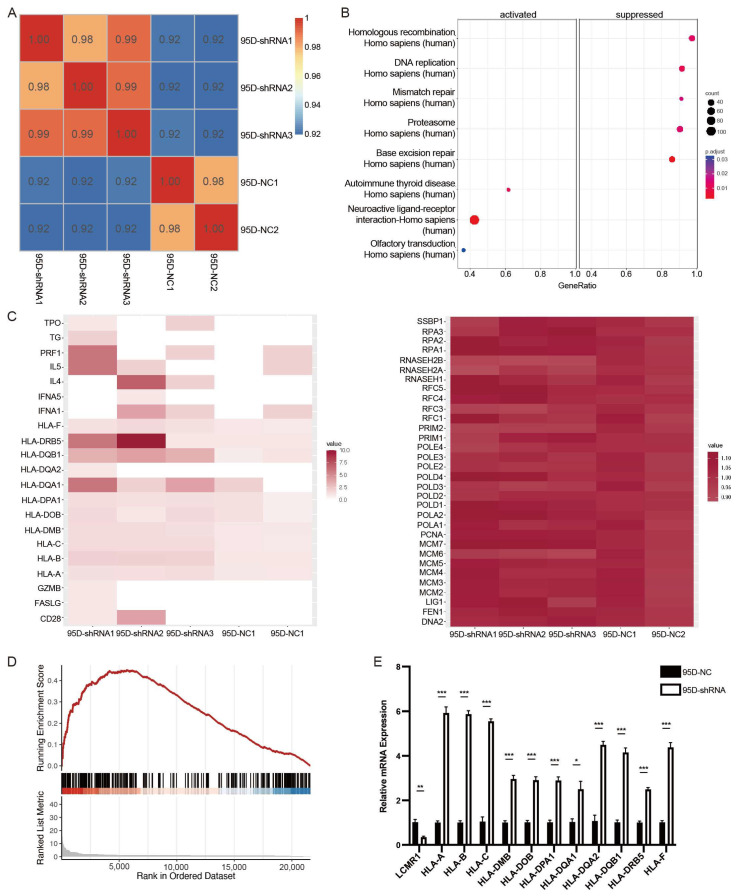
LCMR1 negatively regulates human leukocyte antigen (HLA)-encoding genes (sh-a strain was used for transcriptome sequencing). (**A**) Heat map of the hierarchically clustered Pearson’s correlation matrix resulting from comparing the expression level of each gene in the control and LCMR1-knockdown transcriptomes. (**B**) KEGG pathway enrichment analysis of DEGs. (**C**) Heatmap representation of the expressions of the DEGs associated with autoimmune thyroid disease (left) and homologous recombination (right). (**D**) GSEA plot showing that LCMR1 was inversely correlated with autoimmune thyroid disease signatures. (**E**) Quantitative PCR validation of the upregulation of HLA-encoding genes after LCMR1 knockdown. Results of the qPCR represent the mean and standard deviation of three independent experiments. Student’s *t*-test * *p* < 0.05; ** *p* < 0.01, *** *p* < 0.001.

**Figure 5 cancers-15-05445-f005:**
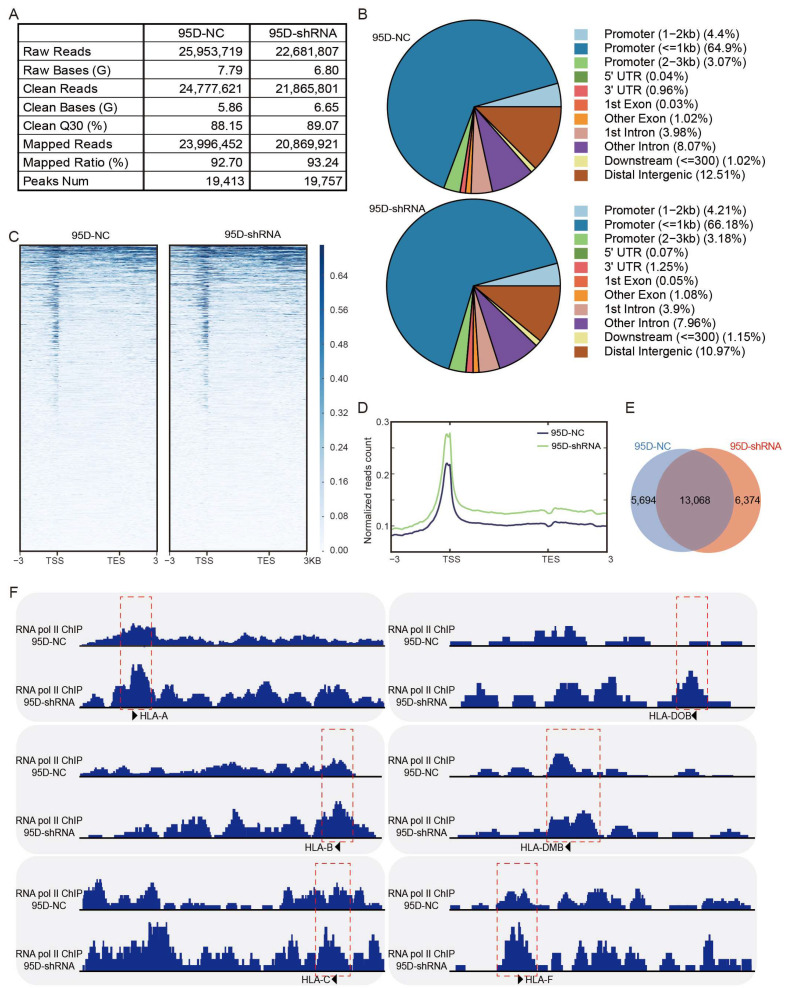
LCMR1 diminishes RNA Pol II occupancy at the promoter of HLA-encoding genes. The stably sh-LCMR1 or sh-NC lentivirus-infected 95D cells were used for RNA Pol II ChIP-seq (Cut & Tag). (**A**) Depth and quality of sequencing. (**B**) Pie chart of the percentages of peak distributions. (**C**) Heatmap displaying all ChIP-seq peaks on the gene body, showing the range from −3 kb upstream of the TSS site to +3 kb downstream of the TES site. (**D**) Normalized read count (average read signals across all genes) across gene body. (**E**) Venn diagram of peaks. (**F**) The IGV tool was used to visualize the binding peaks of RNA Pol II on the promoter region (red dashed box) of Class-I HLA (left) and Class-II HLA genes (right).

**Figure 6 cancers-15-05445-f006:**
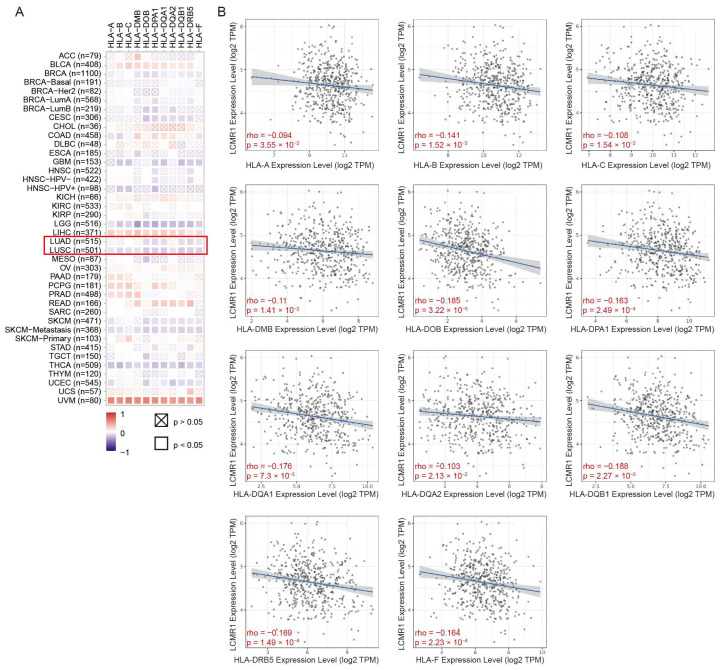
Correlations between LCMR1 mRNA expression and HLA-encoding genes in pan-cancer. (**A**) Heatmap of the correlations between LCMR1 expression and HLA-encoding genes using TIMER2.0. The Z-score method was used to standardize data. Lung adenocarcinoma (LUAD) and lung squamous cell carcinoma (LUSC) are marked by a red box. (**B**) Linear regression correlation of LCMR1 (Y-axis) with each HLA-encoding gene (X-axis) in LUSC.

## Data Availability

The raw datasets, together with the analyzed bigwig and narrowpeak files generated during the current study, are available in the GEO repository GSE 234816.

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
