# Peer review of "LCMR1 Promotes Large-Cell Lung Cancer Proliferation and Metastasis by Downregulating HLA-Encoding Genes"

_cancers, 2023, doi:10.3390/cancers15225445_

Round 1

Reviewer 1 Report

Comments and Suggestions for Authors

Lung cancer is the leading cause of high global morbidity and mortality. NSCLC accounts for about 85% in lung cancer. Lung large-cell carcinoma (LLCC) has the highest aggressive and worst prognosis characteristics in NSCLC. This manuscript revealed that LCMR1 was highly expressed in LLCC 95D cell line. Blocking LCMR1 reduced cell proliferation and metastasis in vitro and in vivo. Mechanistically, LCMR1 inhibited the transcription of HLAs by diminishing the RNA Pol II occupancy at the promoter of HLA coding genes. I have some questions on the manuscript.

1. The aim of study is to analyze the roles of LCMR1 in LLCC. However, there are some discrepancies in the study design. The author analyzed the LCMR1 expression profiles in various tumor tissues based on the TCGA database. Only LUSC and LUAD samples were involved, no LLCC samples. What is the expression pattern of LCMR1 in LLCC samples? Please adding this data in the supplementary files.

2. In in vitro experiments, two lung cancer cell lines 95C and 95D were selected for proliferation and invasion assay. LCMR1 expression was very low in 95C cell line, why do not construct the over-expression vector to confirm the effect of LCMR1 on the proliferation, migration in 95C cells? 

3. Please double check Figure 2C. it is not so clear.

4. In the result section 3.6, “the expression level of LCMR1 negatively correlated with HLA coding genes in NSCLC samples”, thus, author should clarify that the object of study is NSCLC or LLCC.

Comments on the Quality of English Language

Reviewer 2 Report

Comments and Suggestions for Authors

In this article, authors demonstrated LCMR1 expression is high in LCLC cells. Additionally, authors demonstrated that LCMR1 expression is negatively correlated with HLA gene code transcription and LCMR1 promotes LCLC cell proliferation and metastasis. This manuscript is well written. However, there are following concerns:

In figure 1e & 1f authors knocked down LCMR1 expression using shRNA. Then in figure 2 onwards why did they use RNAi. Are authors referring to same shRNA stable knockdown LCMR1 cells throughout the manuscript or are they 2 different types of knockdown? If it is same authors should keep labels same throughout the manuscript. If they are 2 different types of RNA delivery, I do not see this RNAi delivery information in methods section and what is rational for using this when stable knockdown cells were already established.

Authors considered 95C for its low metastatic potential. From figure 1C 95C has slightly lower expression of LCMR1 when compared 95D. So, nominal expression of LCMR1 does not promote metastasis, only high expression promotes metastatic features? or LCMR1 expression itself would drive LCLC cells towards aggressive phenotype? In my opinion authors should consider adding another cell line with slight to no expression of LCMR1 which has non tumorigenic phenotype in figure2.

Figure 2A: Authors should consider adding cell doubling time. Cell doubling time easily shows difference between both cell growth rate. 

Also, authors should consider adding colony formation between 95D-NC and 95D-RNAi. 

Figure 2B: Difference between 95D and 95C and Difference between 95D-NC and 95D-RNAi seems negligible. But, it is surprising to significance between 95D-NC and 95D-RNAi is high whereas no difference in 95D and 95C. 

It is surprising to see cell adhesion % decreased with LCMR1 knockdown. In my opinion floating cells have more metastatic features than adherent cells.  

Figure 2C: It is hard to see cells and scratch lines. Please draw lines on scratch edges. 

Figure 3: As authors are talking about metastasis it would have been beneficial if authors performed this experiment at orthotropic site. Orthotropic would have shown modulation in metastatic potential with LCMR1 expression knockdown. 

Authors should consider adding H&E and IHC of Ki67 or other proliferation marker from harvested tumors. 

Finally, figure 6: Correlation values are pretty weak. It's hard to consider negative correlation from this data. Also, It is also hard to read correlation values. Please increase the font for correlation values on all charts.

Reviewer 3 Report

Comments and Suggestions for Authors

1#. In this work, the authors only knock down LCMR1 in 95D cell line. The evidences were insufficient to draw the conclusion of LCMR1 promotes large-cell lung cancer proliferation and metastasis. They should overexpress LCMR1 in 95C cell line and repeat all phenotypic experiment in vitro and in vivo.

2#. According to literature reports, LCMR1 also promotes proliferation and tumorigenesis of other NSCLC cell lines such as A549 cell. In this work, authors demonstrated that LCMR1 downregulate HLA coding genes in LCLC. Is this mechanism unique to LCLC? I think authors should verify this mechanism in other NSCLC cell lines.

3#. In Figure 1C, there was no control group. The protein level of LCMR1 in BEAS-2B cell should be add as control.  

4#. In Figure 1F, the knockdown effect of only sh-a and sh-e were shown by immunoblotting. It’s better to provide verification for all shRNA constructs. Please indicate that which shRNA constructs was used for transcriptome sequencing in Figure 4.

5#. In Figure 2D, it’s difficult to see obviously difference between all groups. Please repeat the transwell assay or select better representative results.

6#. Please provide the verification of LCMR1 expression by immunoblotting or immunohistochemistry with tumor tissues in Figure 3.

7#. To investigate LCRM1 block the transcription of HLA-coding genes, it’s necessary to perform luciferase reporter assay.

8#. If the authors want to prove LCMR1 promotes large-cell lung cancer proliferation and metastasis by downregulating HLA coding genes, they should find out which HLA coding genes was directly inhibited by LCMR1 and then further knockdown the genes in LCMR1 overexpression cells to see if its downregulation would eliminate LCMR1-induced proliferation and migration or not.

9#. In abstract, the statement of “We fully identified the biofunction of LCMR1 in lung cancer by performing pan-cancer and cell line–based LCMR1 expression profiling” was inappropriate. Since only mRNA level of LCMR1 in pan-cancer was analyzed, not gain or loss of function approaches were performed.

10#. On line 216, the statement of “Profiling data showed that LCMR1 was mainly upregulated in tumor tissues, especially in NSCLC” was inappropriate. The expression of LCMR1 was not obviously high in NSCLC compared to other tumor tissues. It’s better to change to “Profiling data showed that LCMR1 was mainly upregulated in tumor tissues, including NSCLC”.

11#. On line 312, “LCMR1 activates the autoimmune thyroid disease pathway” should be corrected to “LCMR1 depletion activates the autoimmune thyroid disease pathway”.

12#. Please provide more citation about the research of LCMR1 might regulate immune pathway in discussion.

Comments on the Quality of English Language

The presentation is good and easy to understand. 

Round 2

Reviewer 1 Report

Comments and Suggestions for Authors

Thank you for your all response to my comments. The revised version of manuscript is qualified for the acceptance of this journal.

Reviewer 3 Report

Comments and Suggestions for Authors

The authors supplemented the experiments as suggested and answered all questions carefully. The article content has been corrected and the language improved.